# Evaluation of MELD Scores and Thyroid Hormones as Prognostic Factors of Liver Cirrhosis

**DOI:** 10.3390/medicina60091474

**Published:** 2024-09-09

**Authors:** Anca M. Belu, Alina D. Nicoara, Daniela M. Belu, Eduard Circo

**Affiliations:** 1Faculty of Medicine, Ovidius University of Constanta, 1 Universitatii Alley, 900470 Constanta, Romania; ancabelu@doctor.com (A.M.B.); belu.d@yahoo.com (D.M.B.); eduard_circo@yahoo.com (E.C.); 2“St. Apostol Andrew” Emergency County Hospital, 145 Tomis Blvd., 900591 Constanta, Romania

**Keywords:** evolution of cirrhosis, TSH, T3, fT4, encephalopathy, liver

## Abstract

*Background and Objectives*: Hepatic cirrhosis is a disease with an increasing frequency globally, but its mechanisms of disease development are not yet completely known. The aim of this study was to evaluate the relationship between thyroid hormone levels (T3, fT4, and TSH) and survival in patients with chronic liver disease. *Materials and Methods*: A total of 419 patients diagnosed with liver cirrhosis were included in the study. The MELD score was computed, and TSH, T3, and fT4 were collected from each patient using the ELISA procedure. Signs and symptoms of liver failure and portal hypertension confirmed the clinical diagnosis of liver cirrhosis, and biological tests and imaging methods confirmed the diagnosis. *Results*: The MELD score was positively associated with TSH on admission and TSH on discharge and negatively associated with T3 at discharge. TSH levels were higher in non-survivors than in survivors. The values of T3 and fT4 present no significant changes to be considered as prognostic factors. *Conclusions*: Although the differences between the median TSH values of the patients who died and those who survived are not very large, the statistical significance of the data obtained demonstrates that there are changes in metabolism of the thyroid hormones during the progression of liver cirrhosis. It is possible that TSH is the one which maintains the normal balance of thyroid activity for patients with liver cirrhosis, so it can be considered as an important marker of evolution of these patients.

## 1. Introduction

Liver cirrhosis has become an increasingly public health concern globally and in Europe. The Global Burden of Disease (GBD) study (2017) estimated that 112 million people worldwide have been diagnosed with compensated liver cirrhosis [1]. According to WHO, 2.4% of all deaths worldwide are due to liver cirrhosis [2].

The main causes of liver cirrhosis are infection with hepatitis C virus (HCV), hepatitis B virus (HBV), alcohol-associated liver disease, and non-alcoholic fatty liver disease (NAFLD) [3].

There is a tendency for viral liver infections to decrease as the cause of liver cirrhosis. This fact is due to the vaccination against virus B and the implementation of treatment programs for HBV, but also the successful treatment of HCV infection with direct acting antiviral (DAA) therapy [4].

Alcohol is the substance most often abused throughout the world and is still a main cause in the etiology of liver cirrhosis [5].

Alcohol consumption is an important factor in the occurrence and evolution of liver cirrhosis. The systemic effect of alcohol consumption causes complications of pre-existing pathologies, but also the unfavorable evolution of liver pathology. The association between viral liver infection and alcohol consumption causes repeated decompensations of liver disease and forms with unfavorable evolution [6].

End-stage chronic liver disease is characterized by the replacement of normal liver tissue with fibrotic tissue. Liver function loss has serious repercussions and is a cause of morbidity and mortality [7].

The management of chronic liver diseases is challenging because of the complexity of liver functions and the metabolic correlations in which they play a decisive role [8].

The survival of patients with liver cirrhosis depends on the way in which the triggering cause of cirrhosis is kept under control, but also the complications and the effect on other associated pathologies.

Liver cirrhosis develops over time and can have permanent, potentially fatal consequences. Monitoring the clinical course and treatment is challenging because of the lack of prognostic and developmental markers. The evolution of patients with end-stage chronic liver disease is completely unpredictable because of the lack of balance between cell destruction and regeneration and the absence of a clear link between numerous biological constants and disease progression [9]. Challenging research topics include the prevention of these problems and the determination of the links between disease progression and metabolic changes, including hormonal changes [10]. Recent research has traced thyroid dysfunction to liver disease. Changes in thyroid-stimulating hormone (TSH) and thyroid hormone values are associated with liver disease complications, including mortality [11,12].

Thyroid hormone profile was strongly associated with worse outcomes in patients with cirrhosis and might represent a promising prognostic tool that can be incorporated in clinical practice [13].

The thyroid hormones triiodothyronine (T3) and thyroxine (T4) are produced under the control of an endocrine feedback loop. Both hormones are bound to the bloodstream by transport proteins called albumin, transthyretin, and thyroid-binding globulin (TBG) [14]. They play a part in the proper growth, development, and operation of organs. Moreover, they influence liver function by regulating the basal metabolic rate of all cells, including hepatocytes. On the other hand, the liver plays a significant role in thyroid hormone metabolism, including conjugation, thyroglobulin-related synthesis, and peripheral deionization [15]. Triiodothyronine is the main regulator of thyroid function in various target organs. Most of the T3 hormone is produced by enzymatic deiodination at the 5′ position of T4, mainly in the liver. Thus, T3 reflects the functional status of peripheral tissue rather than synthetic thyroid activity. Serum T3 concentration decreases as the conversion of T4 to T3 decreases [16].

The activity of the thyroid gland is directly connected with that of the liver. Thyroid hormones regulate the rate of basal metabolism of hepatocytes and dysthyroidism can produce alterations in liver metabolism and circulation at this level [17]. Thyroid hormones elicit non-genomic effects that usually begin at the plasma membrane and are mediated primarily by integrin αvβ3, although other receptors such as TRα and TRβ are also capable of eliciting non-genomic responses [18].

There is a complex relationship between thyroid and liver pathology. Under normal conditions, the liver plays an essential physiological role in thyroid hormone activation and inactivation, transport, and metabolism, while thyroid hormones are involved in hepatocyte activity and liver metabolism [19]. In hypothyroidism, changes in liver enzymes can appear, which can be attributed to the impairment of lipid and protein metabolism. Also, severe hypothyroidism can develop with hyperammonemia and ascites, mimicking liver failure [20].

Thyroid hormones participate actively or inactively in all physiological processes in the body [21]. The homeostasis of the thyroid can affect the evolution of chronic liver diseases; there is even the idea that use of thyroid hormone-analog can be used in the treatment of liver disease [22].

Patients with liver cirrhosis may be clinically euthyroid, but the determination of thyroid hormone and TSH values may record changes most likely determined by metabolic alteration, especially related to dysproteinemia, with a decrease in total proteins and albumin [23].

The MELD score is commonly used to assess disease severity and it is based on paraclinical measurements: bilirubin, creatinine, and INR (International Normalized Ratio). We studied the possibility that calculating and tracking the MELD score and detecting changes in the level of thyroid hormones may improve the evolution of the patients with liver cirrhosis [24,25].

The main aim of this study was to investigate the possibility of using combined thyroid hormones and the MELD score to more accurately evaluate the mortality of patients with chronic liver disease. In this regard, as a specific objective, we aimed to evaluate the relationship between thyroid hormone levels (T3, fT4, and TSH) and the survival of patients with chronic liver disease.

## 2. Materials and Methods

Survival was assessed by recording death during hospitalization in patients with chronic liver disease. MELD (Model for End-Stage Liver Disease) is a reliable indicator of short-term survival in patients with end-stage liver disease and was designed based on bilirubin, creatinine, and INR. The lowest MELD score was 6, and the highest score was 40.

### 2.1. Samples

Thyroid Stimulating Hormone (TSH, thyrotropin): collection was performed under fasting conditions, but not after a recent thyroid biopsy or thyroid surgery. Venous blood was sampled using a collection container with a vacutainer without anticoagulant, with or without separating gel. Serum storage conditions: 20 °C or 2–8 °C. The samples were centrifuged to separate it. At least 0.5 mL of serum was extracted, and electrochemiluminescence detection immunochemistry (ELISA) was used as the analytical method. We excluded any analytical or drug interference and evaluated the range between 0.27 and 4.2 IU/mL as normal values, taking into account the adult age of the patients in the research group [26].

Thyroxine (T4): Blood was collected from venous blood in a vacutainer with or without separating gel and without anticoagulant. The storage conditions were 20 °C or 2–8 °C. Centrifugation was used to separate the serum. The serum sample should contain 0.5 mL. The electrochemiluminescence detection immunochemical assay (ELISA) was performed. Interferences in drug and kit components were excluded. Reference values ranged from 12.0 to 22.0 pmol/L [27].

Triiodothyronine (T3) is a thyroid hormone that circulates bound to the transporter protein but has a 10-fold lower affinity for the protein transporter than T4. The collection and determination method was similar to that for fT4 determination, and the reference range is 1.3–3.1 nmol/L [16].

All serological samples were processed during hospitalization and in the same laboratory.

### 2.2. Inclusion Criteria

The study included all patients over 18 years of age who provided written informed consent and were diagnosed with liver cirrhosis after clinical, paraclinical-biologic, and imaging examinations.

### 2.3. Exclusion Criteria

Patients who refused to complete the informed consent form were excluded. Patients who were taking drugs that might have affected thyroid hormone metabolism were also excluded. These included iodide contrast agents and amiodarone, which inhibit the conversion of T4 to triiodothyronine T3, as well as other classes of drugs, such as glucocorticoids and dopamine, which increase TSH secretion and, therefore, decrease T3 [28].

### 2.4. Participants

The study involved 419 patients diagnosed with liver cirrhosis who were hospitalized between March 2022 and March 2023 in Constanța County Hospital, Romania. The patients were followed during hospitalization, until discharge, with an interval between 1 and 41 days (M = 7.21, SD = 6.73). There were three categories of discharges—clinical and biological stabilization with resolution of the decompensation episode, discharge on request, and exitus.

### 2.5. Data Analysis Methods

All analysis were performed using R [29] and the R packages.

Initially, outliers, missing values, and univariate descriptive analyses, including analysis of compliance with the assumption of univariate normality for continuous data, were performed using the Shapiro–Wilk test statistic [30,31], and indicators of skewness and kurtosis were computed. Extreme univariate values (located beyond 3 standard deviations to the left or above 3 standard deviations to the right of the mean) were replaced using missing values, and imputation was performed using the K-nearest neighbor method [32]. Recorded hormone values as they resulted from the analysis (continuous variables, higher power) were used instead of transformation to categorical values (categorical variables, lower power), and statistical methods were used to compare rank means.

For the analysis of the associations between the main continuous variables, depending on whether the assumption of univariate normality was met, a Bravais–Pearson r correlation matrix was used if the assumption was met, or a Spearman ρ correlation matrix was used if the assumption was not met.

To test the hypotheses, depending on the fulfillment of the assumptions, either the two-sample *t*-test, comparing the means of two independent populations, or its nonparametric equivalent (Wilcoxon sum rank test) was used.

Given that the survival achievement variable is dichotomous, its prediction as a function of thyroid hormone levels was performed using receiver operator characteristic (ROC) curve and area under curve (AUC) analysis. The true positive rate (TPR) was calculated as the proportion of correctly predicted survival cases out of all survival cases, and the false positive rate (FPR or specificity) as the proportion of incorrectly predicted cases out of all deaths. The ROC curve displays the ratio between sensitivity and specificity, and closeness to the upper-left corner indicates very good classification performance, whereas the equality of sensitivity and specificity (TPR = FPR) indicates lack of concrete classification of predictions (random assignment), where the ROC curve follows the diagonal. AUC (Area Under Curve) and confidence interval were also calculated as measures of classification accuracy. The control level for the ROC curve analysis was death (the “Yes” variant).

The diagnosis of liver cirrhosis was supported clinically by the presence of portal hypertension and symptoms of liver failure and paraclinical findings on biological and imaging tests. Each patient was assigned a MELD score and a Child–Pugh score, depending on the course and severity. We also classified the patients with hepatic encephalopathy into stage categories, using the traditional West Haven Classification (formulated by Harold Conn). This classification divides patients with hepatic encephalopathy into four stages [33]. According to the West Haven criteria, stage 1 includes changes in attention, euphoria, or anxiety, and reduced intellectual performance; stage 2 is characterized by lethargy or apathy, minimal temporo-spatial disorientation, personality changes, and inappropriate behavior; stage 3 includes drowsiness up to semi-stupor, but remaining responsive to verbal stimuli, confusion, and severe temporal-spatial disorientation; and stage 4 is represented by coma [34].

Kaplan–Meier survival analysis was also conducted, in which participants were right-censored according to the number of hospitalization days. As an event of interest, the death of patients was monitored, and as a group variable, continuous hormone values were discretized, as follows: (a) for TSH, values between 0.27 and 4.2 were considered normal, and values lower or higher than these limits were considered abnormal; (b) for T3, values between 1.3 and 3.1 were considered normal, and values lower or higher than these limits were considered abnormal; and (c) for fT4, values between 12.0 and 22.0 were considered normal, and values lower or higher than these limits were considered abnormal.

## 3. Results

### 3.1. Socio-Demographic Data

The age of the participants ranged from 32 to 91 years (M = 63.26, SD = 9.57), and 68.97% were male. After diagnosis, patients were hospitalized between 0 and 96 months (M = 20.35, SD = 20.22), and the duration of hospitalization ranged from 0 to 41 days (M = 7.21, SD = 6.73).

In terms of etiology, most cases had alcoholic etiology (50.60%), followed by HVC (18.62%) and HVB (17.90%) etiology, as well as mixed etiology (12.89%). In terms of encephalopathy score, most patients were in the group determined by score 0 (67.78%), followed by those in the group determined by a score of 2 (17.66%), then those in the group with score 1 (6.68%) and 3 (5.01%), and lastly those with the highest score, 5 (2.86%). Moreover, (7.88%) are currently deceased.

### 3.2. Univariate Descriptive Analysis

Values > 10.4 for TSH at discharge, >5.4 for T3 at admission, and >4.3 for T3 at discharge were considered univariate extreme and were discarded, with a new K-nearest neighbor imputation. For TSH, only one missing value was found at admission and three missing values at discharge. For T3, five missing values were observed, the same participants at admission and discharge. For FT4, two missing values were found at admission and three missing values at discharge

The hospitalization data showed that the MELD score, TSH, T3, and fT4 are positively skewed, postulating the existence of highly, although not extremely, emphasized values. The MELD score and TSH have a leptokurtic distribution, with low variability around the mean, and the other variables had a mesokurtic distribution (Table 1 and Table 2), so the assumption of univariate normal distribution is not fulfilled.

In the case of the discharge data, all distributions are positively skewed, and the distribution for T3 is platykurtic (see Table 1).

### 3.3. Bivariate Correlation Analysis

The assumption of univariate normality was not met, so the variables were correlated using Spearman’s ρ correlation coefficient (Table 3) after first converting the dichotomous variable “non-surviving patients” to the corresponding numerical values.

The MELD score was positively associated with TSH on admission (ρ = 0.27, *p* < 0.001), and TSH on discharge (ρ = 0.21, *p* < 0.001), negatively associated with T3 at discharge (ρ = −0.17, *p* < 0.001), negatively marginally associated with non-survival (ρ = −0.09, *p* = 0.074) and T3 on admission (ρ = −0.10, *p* = 0.053), and not associated with hospitalization days (ρ = −0.03, *p* = 0.541), fT4 at hospitalization (ρ = −0.02, *p* = 0.629), and fT4 at discharge (ρ = 0.08, *p* = 0.109).

Non-survivor status at discharge was negatively associated with TSH at hospitalization (ρ = −0.19, *p* < 0.001), and TSH at discharge (ρ = −0.14, *p* = 0.004), and not associated with hospitalization days (ρ = −0.02, *p* = 0.702), T3 at hospitalization (ρ = 0.04, *p* = 0.0355), T3 at discharge (ρ = −0.02, *p* = 0.750), fT4 at hospitalization (ρ = 0.02, *p* = 0.745), and fT4 at discharge (ρ = 0.02, *p* = 0.745).

The number of hospitalization days was not associated with any variable.

TSH on admission was positively associated with TSH on discharge (ρ = 0.75, *p* < 0.001), negatively associated with T3 on admission (ρ = −0.35, *p* < 0.001), T3 on discharge (ρ = −0.23, *p* < 0.001), and fT4 on admission (ρ = −0.18, *p* < 0.001), and not associated with fT4 on discharge (ρ = 0.05, *p* = 0.315).

TSH at discharge was positively associated with fT4 at discharge (ρ = 0.20, *p* < 0.001), negatively associated with T3 at hospitalization (ρ = −0.28, *p* < 0.001), and T3 at discharge (ρ = −0.11, *p* = 0.005), and not associated with fT4 at hospitalization (ρ = −0.07, *p* = 0.147).

T3 on admission was positively associated with T3 on discharge (ρ = 0.46, *p* < 0.001), fT4 on admission (ρ = 0.23, *p* < 0.001), and marginally positively associated with fT4 on discharge (ρ = 0.09, *p* = 0.08).

T3 at discharge was positively associated with fT4 at hospitalization (ρ = 0.13, *p* = 0.008) and fT4 at discharge (ρ = 0.50, *p* < 0.001), and fT4 at hospitalization was positively associated with fT4 at discharge (ρ = 0.42, *p* < 0.001).

### 3.4. Data Analysis

The non-parametric Wilcoxon rank sum test was used for two independent populations from which the samples were drawn, also known as the Mann–Whitney test, because the assumption of univariate normality was not met.

We found that the mean value for TSH on admission was 7.14 (SD = 1.79, median = 6.7) in decedents and 5.78 (SD = 1.9, median = 6.60) in survivors, with normal values ranging from 0.4 to 4.0 mIU/L (Figure 1). Thus, the decedents had statistically significantly higher TSH levels on admission (W = 8930, *p* < 0.001, es = 0.19), and the effect size was small.

The space between the dotted lines is the space of normal range of values. Similar results were also observed for TSH at discharge, the mean rank of survivors’ values being 3.85 (SD = 1.26, median = 3.55), lower compared to that of the deceased 4.31 (SD = 1.19, median = 3.9), and this difference had statistical significance (W = 8260, *p* = 0.005, es = 0.14) with a small effect size.

ROC analysis showed an accuracy of survival classification based on TSH hormone of 70.33% (95% CI [61.61, 78.05]) at admission, and (62.78%, 95% CI [52.86, 71.27]) at discharge (Figure 2), so it can be considered an acceptable classifier of survival in patients with cirrhosis, even though the effects have been shown to be very small but statistically significant.

In the case of T3 on admission, the values of the decedents were (M = 0.74, SD = 0.38, median = 0.76), and of the survivors were (M = 0.85, SD = 0.50, median = 0.77), with no statistically significant differences (W = 5750.5, *p* = 0.354, es = 0.05) between the two means (Figure 3).

The space between the dotted lines is corresponding to the normal values of T3. At discharge, those who subsequently died had a mean of 0.77 (SD = 0.47, median = 0.80), and the mean of those who survived was 0.78 (SD = 0.53, median = 0.66), with at most marginally significant differences between the two means (W = 6581.5, *p* = 0.75, es = 0.02).

Survival classification based on T3 hormone had a very low accuracy both at admission 55.95% (95% CI [46.41, 64.74]) and at discharge, with a specificity of 50.59% (95% CI [41.18, 59.87]) at discharge (Figure 4), and T3 cannot be considered an acceptable classifier of survival in patients with liver cirrhosis.

For fT4, there were no statistically significant differences between survivors and those subsequently deceased, neither at admission (W = 6229.5, *p* = 0.835, es = 0.01) nor at discharge (W = 6155.5, *p* = 0.745, es = 0.02).

Classification of survival based on fT4 had a very low accuracy, both at admission 52.54% (95% CI [41.77, 60.94]) and at discharge (52.07%, 95% CI [41.08, 64.29]). Therefore, fT4 cannot be considered an acceptable classifier of survival in patients with liver cirrhosis.

The mean MELD values were 15.54 (SD = 5.20, median = 15) for decedents and 13.95 (SD = 4.22, median = 14) for survivors. Thus, deceased patients have marginally significantly higher MELD score values (W = 7555.5, *p* = 0.075, es = 0.09), but the effect size was small.

Kaplan–Meier analysis was performed only for admission data. In the case of TSH, the average number of days of hospitalization was 7.34 (N = 327, SD = 6.88) for patients with abnormal hormone values and 6.74 (N = 92, SD = 6.16) for patients with normal values, with patient censoring observed almost every day. Our data show that all patients with normal TSH values survived, while for patients with abnormal TSH values the probability of survival was 88.8% (SE = 2.39%, 95% CI [84.2%, 93.6%]) at 9 days and 46.6% (SE = 19.52%, 95% CI [20.5%, 100.0%]) at 35 days. The log-rank test indicates a statistically significant difference (*p* = 0.003). The median survival time of patients in the group with abnormal TSH values was 35 days, which is significantly shorter than the survival time of patients in the group with normal TSH values (41 days, see Figure 6).

In the case of T3 values, the average number of days of hospitalization was 7.29 (N = 328, SD = 6.77) for patients with abnormal hormone values and 6.90 (N = 91, SD = 6.61) for patients with normal values, with patient censoring observed almost every day. Our data show that for patients with abnormal T3 values the probability of survival was 89.0% (SE = 24.16%, 95% CI [84.3%, 93.9%]) at 9 days and 71.5% (SE = 6.46%, 95% CI [59.9%, 85.4%]) at 22 days, whereas for patients with normal T3 values it was 92.3% (SE = 6.45%, 95% CI [80.5%, 100.0%]) at 12 days (Figure 5). The log-rank test indicated no statistically significant difference (*p* = 0.10) between the median survival time of patients with abnormal T3 values and that of patients with normal T3 values.

The same conclusions can be drawn for fT4. The average number of days of hospitalization was 7.78 (N = 97, SD = 7.40) for patients with abnormal hormone values and 7.03 (N = 322, SD = 6.52) for patients with normal values, with patient censoring observed almost every day. For patients with abnormal fT4 values, the probability of survival was 90.40% (SE = 5.03%, 95% CI [81.0%, 100.0%]) at 12 days and 73.8% (SE = 11.84%, 95% CI [53.9%, 100.0%]) at 22 days, whereas for patients with normal fT4 values it was 85.3% (SE = 3.39%, 95% CI [79.0%, 92.3%]) at 12 days and 77.5% (SE = 5.37%, 95% CI [67.7%, 88.8%]) at 18 days (Figure 6). The log-rank test indicated no significant difference (*p* = 0.93) between the median survival time of patients with abnormal fT4 values and that of patients with normal fT4 values.

## 4. Discussion

MELD score was positively associated with TSH at admission and at discharge and negatively associated with T3 at discharge. This finding corresponds with data published by Punekar, but refuted by studies by Vincken [11,35].

Non-survival was negatively associated with TSH at admission and at discharge and not associated with the number of hospitalizations, not associated with T3 at admission and at discharge, and not associated with fT4 at admission and at discharge. The number of days of hospitalization was not associated with any of the variables.

TSH at admission was positively associated with TSH at discharge, negatively associated with T3 at admission and at discharge and with fT4 at admission, and not associated with fT4 at discharge. The same situation is presented in the study of Vinken [35].

T3 at discharge was positively associated with fT4 at admission and at discharge. The deceased had statistically significantly higher TSH levels on admission and also at discharge, so it can be considered an acceptable classifier of survival of patients with liver cirrhosis. These results are consistent with the data obtained by Fei Ye, who considered the association between increased TSH and mortality to be statistically significant [36].

There are no statistically significant differences between T3 on admission to the hospital of the survivors and of the decedents, so T3 cannot be considered as an acceptable classifier of survival in liver cirrhosis. The same thing can be said about fT4, and the result is in agreement with studies by Trajkovic and Vincken who reported that they observed no associations between fT4 values and cirrhotic patients who died [35,37].

The Kaplan–Meier analysis shows that all the patients with normal TSH survived and for those with abnormal TSH values the probability of survival was 88.8% at 9 days of hospitalization and 46.6% at 35 days. The log-rank test shows no statistically significant difference in T3 and fT4 in survivors and in non-survivors. A similar situation was reported in the studies of Fei Ye [36]. Although the differences between the median TSH values of the patients who died and those who survived are not very large, the statistical significance of the data obtained demonstrates that there are changes in metabolism of the thyroid hormones during the progression of liver cirrhosis.

For fT4, there were no statistically significant differences between survivors and those who subsequently died, and the result is in agreement with studies by Trajkovic and Vincken who reported that they observed the same situation [35,37].

## 5. Conclusions

The findings of this study highlight the importance of careful monitoring of hormonal markers in patients with liver cirrhosis. The MELD score was positively associated with TSH on admission and TSH on discharge, and negatively associated with T3 at discharge. Increased TSH levels in patients with cirrhosis during hospitalization are associated with mortality. TSH may be a prognostic factor of mortality in patients with liver cirrhosis. Monitoring TSH may not only improve our understanding of disease progression, but significantly contribute to patient survival. The MELD score provides important prognostic and evolution data. These data can be correlated with the TSH variation and can be used to prevent unfavorable evolution towards exitus. The T3 and fT4 changes that occur during the decompensation of patients with liver cirrhosis can be considered transitory and not part of the permanent damage to the thyroid gland. The role of TSH is major, correcting thyroid hormone disorders that can occur in patients with liver cirrhosis. Even the conversion from T4 to T3 may not occur in proper conditions due to impaired liver function; it seems that the body has the ability to regulate the secretion of thyroid hormones so that there are no significant changes. This is probably achieved through the effect of TSH, which shows increased values along with the evolution towards death of patients with liver cirrhosis.

## Figures and Tables

**Figure 1 medicina-60-01474-f001:**
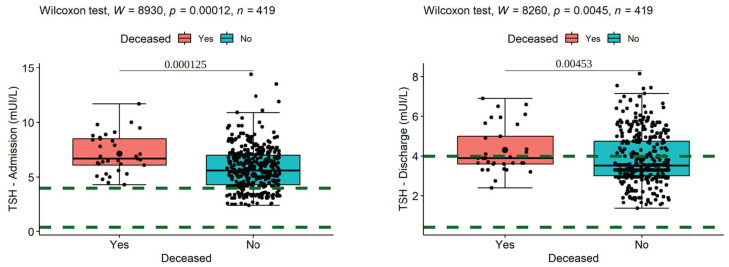
Comparisons of mean ranks of deceased and survivors’ TSH levels on admission (**left**) and discharge (**right**).

**Figure 2 medicina-60-01474-f002:**
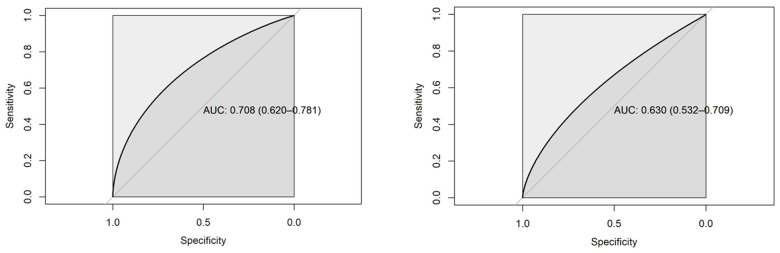
ROC curve analysis of survival classification according to TSH on admission (**left**) and discharge (**right**).

**Figure 3 medicina-60-01474-f003:**
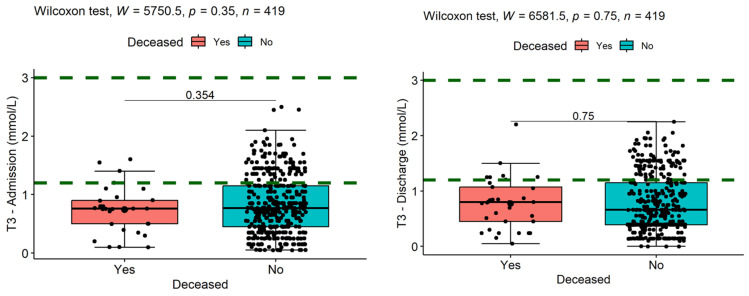
Comparisons of mean ranks of deceased and survivor T3 values on admission (**left**) and discharge (**right**).

**Figure 4 medicina-60-01474-f004:**
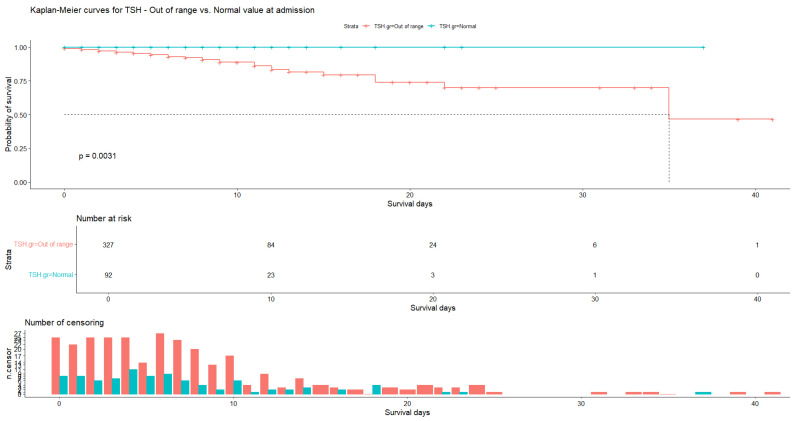
Kaplan–Meier curve for TSH at admission. Normal values vs. out-of-range values.

**Figure 5 medicina-60-01474-f005:**
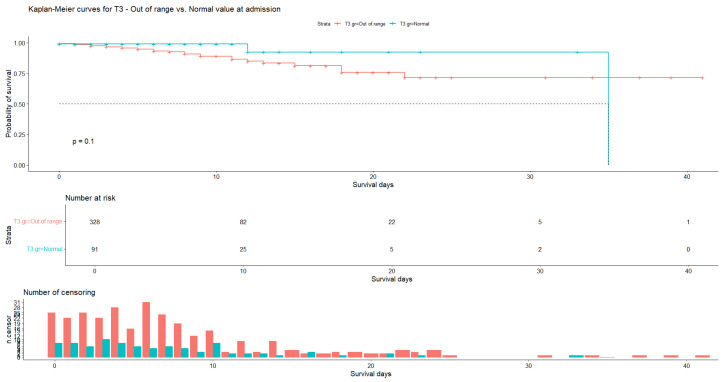
Kaplan–Meier curve for T3 at admission. Normal values vs. out-of-range values.

**Figure 6 medicina-60-01474-f006:**
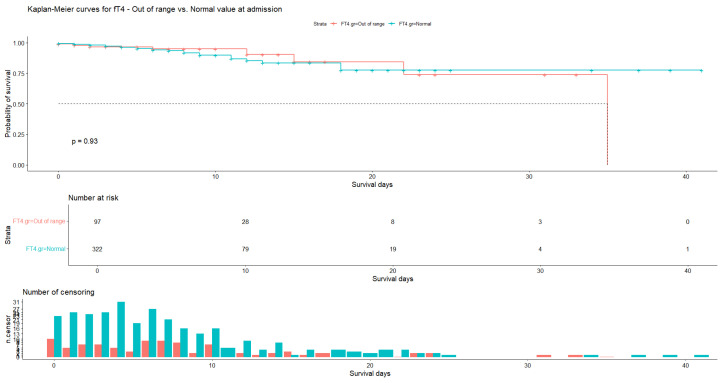
Kaplan–Meier curve for fT4 at admission. Normal values vs. out-of-range values.

**Table 1 medicina-60-01474-t001:** Univariate descriptive analysis: data collected at admission.

Variable	N	Mean	Ab. Std	Median	Min	Max	Skew (ES)	Kurt (ES)	Normal
MELD Score	419	14.08	4.32	14	5	31	0.72 (0.12)	0.92 (0.24)	
TSH (mUI/L)	419	5.89	1.92	5.7	2.4	14.4	0.70 (0.12)	1.03 (0.24)	0.4–4.0
T3 (pmol/L)	419	0.84	0.50	0.77	0.05	2.5	0.51 (0.12)	−0.24 (0.24)	1.2–3.0
fT4 (pmol/L)	419	14.13	2.62	13.5	8.8	23	0.83 (0.12)	0.43 (0.24)	12.0–22.0

**Table 2 medicina-60-01474-t002:** Univariate descriptive analysis: data collected at discharge.

Variable	N	Mean	Ab. Std	Median	Min	Max	Skew (ES)	Kurt (ES)	Normal
TSH (mUI/L)	419	3.89	1.26	3.55	1.35	8.15	0.68 (0.12)	0.04 (0.24)	0.4–4.0
T3 (pmol/L)	419	0.78	0.52	0.65	0.00	2.25	0.64 (0.12)	−0.64 (0.24)	1.2–3.0
fT4 (pmol/L)	419	12.76	3.04	12.4	8.8	23.4	0.86 (0.12)	0.26 (0.24)	12.0–22.0

**Table 3 medicina-60-01474-t003:** Spearman ρ correlation matrix.

	1	2	3	4	5	6	7	8	9
(1) MELD	-								
(2) Non-surviving patients	−0.09	-							
(3) Hospitalization (days)	−0.03	−0.02	-						
(4) TSH—Admission	0.27 ***	−0.19 ***	0.03	-					
(5) TSH—Discharge	0.21 ***	−0.14 **	0.00	0.75 ***	-				
(6) T3—Admission	−0.10	0.04	0.01	−0.35 ***	−0.28 ***	-			
(7) T3—Discharge	−0.17 ***	−0.02	−0.01	−0.23 ***	−0.14 **	0.46 ***	-		
(8) fT4—Admission	−0.02	0.01	0.06	−0.18 ***	−0.07	0.23 ***	0.13 **	-	
(9) fT4—Discharge	0.08	0.02	0.07	0.05	0.20 ***	0.09	0.51 ***	0.42 ***	-
Media	14.08	1.92	7.21	5.89	3.89	0.84	0.78	14.13	12.76
Standard deviations	4.32	0.27	6.73	1.92	1.26	0.50	0.52	2.62	3.04

*** *p* < 0.001; ** *p* < 0.01.

## Data Availability

Data are available on request.

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
