# Peer review of "Evaluation of MELD Scores and Thyroid Hormones as Prognostic Factors of Liver Cirrhosis"

_medicina, 2024, doi:10.3390/medicina60091474_

Round 1

Reviewer 1 Report

Comments and Suggestions for Authors

The manuscript presents an interesting original research article on the association of thyroid status and liver cirrhosis prognosis, however, there are a multitude of issues I would like to address:

1.  In the first sentence of the introduction (line 35) there is a repetition of the phrase "a public health concern" - this must be corrected.

2. The introduction on chronic liver diseases sounds too vague and oversimplistic, I recommend expanding this paragraph with more details. Most of the sentences in this section use basic terms unsuitable for a scientific article.

3. In line 65 there is a repetition of similar ideas - "mainly in the liver, occurs outside the thyroid"

4. In line 66 "thyroid activity" should be changed to "thyroid synthetic activity" for better understanding

5. In line 70 what is "the question" referring to? It should be changed to "a".

6. What does "liver disease" mean in line 81 as it is not a component of the MELD scoring system? 

7. I would like to inquire in what language is the phrase "a jeune people's conditions" in line 85 because it is currently incomprehensible in English.

8. There is a misconception in line 99 - T3 is not only a binding globulin by itself, but a compound of the free hormone and its binding proteins. Also what does the 10-fold lower affinity refer to? Please clarify the meaning of the sentece. 

9. There is a crucial factual error in line 109 as amiodarone does not enhance the conversion of fT4 to fT3, but on the contrary, inhibits. The same applies to the example of glucocorticoids. Those errors imply a critical lack of knowledge of endocrinology principles.

10. In line 152 the abbreviation "TE" is used without prior mentioning of its meaning.

11. The results despite being statistically significant are of dubious clinical significance, as both groups of alive and deceased patients had TSH values in the euthyroid range, with their medians being decimal values apart. I would like the authors to define what is the clinical significance of those findings.

12. Even if the results are considered statistically and clinically significant, a plausable mechanism of the correlation between thyroid status and liver cirrhosis mortality should be discussed.

13. As of the above, even if the results are considered significant, is there a proposed intervention to influence the course of the disease, as both groups had their median thyroid hormone levels in the eurthyroid range?

Comments on the Quality of English Language

There are some serious flaws with the English language used throughout the manuscript such as repetitive phrases, short and oversimplistic sentences, incorrect and vague terms, so I recommend a major revision of the text presented.

Author Response

Thank you for giving me the opportunity to submit a revised draft of my manuscript titled “Evaluation of MELD score and thyroid hormones as prognostic factors in liver cirrhosis”   to     Medicina.  I appreciate the time and effort that you have dedicated to providing your valuable feedback on my manuscript.

Comments 1- In the first sentence of the introduction (line 35) there is a repetition of the phrase "a public health concern" - this must be corrected.

Response 1- Thank you for your good observation. It is done in the text.

Comments 2- The introduction on chronic liver diseases sounds too vague and oversimplistic, I recommend expanding this paragraph with more details. Most of the sentences in this section use basic terms unsuitable for a scientific article.

Response 2- We agree with this and  we have incorporated your suggestion throught the manuscript.

Comments 3-   In line 65 there is a repetition of similar ideas - "mainly in the liver, occurs outside the thyroid"

Response 3- We agree,  the correction is made.

Comments 4- In line 66 "thyroid activity" should be changed to "thyroid synthetic activity" for better understanding

Response 4 – It is done, thank you.

               Comments 5-  In line 70 what is "the question" referring to? It should be changed to "a".

Response 5 – We agree, that it was not the best formulation. It is fixed.

   Comments 6- What does "liver disease" mean in line 81 as it is not a component of the MELD scoring system? 

               Response 6- Thank you for pointing this out. We have done the rectification.

Comments 7-  I would like to inquire in what language the phrase "a jeune people's conditions" is in line 85 because it is currently incomprehensible in English.

Response 7 – You are right; we used the French term, which wasn’t proper. It is fixed.

Comments 8- There is a misconception in line 99 - T3 is not only a binding globulin by itself but a compound of the free hormone and its binding proteins. Also, what does the 10-fold lower affinity refer to? Please clarify the meaning of the sentence. 

Response 8 – We have clarified the meaning of the sentence.

Comments 9 - There is a crucial factual error in line 109 as amiodarone does not enhance the conversion of fT4 to fT3, but on the contrary, inhibits it. The same applies to the example of glucocorticoids. Those errors imply a critical lack of knowledge of endocrinology principles.

Response 9-  We agree with these comments, we fixed all the errors.

Comments 10- In line 152 the abbreviation "TE" is used without prior mentioning of its meaning.

Response 10- We made the change, it was about alcoholic etiology (toxic-ethanolic).

Comments 11- The results despite being statistically significant are of dubious clinical significance, as both groups of alive and deceased patients had TSH values in the euthyroid range, with their medians being decimal values apart. I would like the authors to define what is the clinical significance of those findings.

Response 11 - Thank you for pointing this out. We explained the obtained results more clearly and insisted more on the clinical value of the obtained data.

Comments 12 - Even if the results are considered statistically and clinically significant, a plausible mechanism of the correlation between thyroid status and liver cirrhosis mortality should be discussed.

Response 12- Considering this observation, we revised the discussions for a better explanation of the results obtained.

Comments 13- As of the above, even if the results are considered significant, is there a proposed intervention to influence the course of the disease, as both groups had their median thyroid hormone levels in the euthyroid range?

Response 13- We revised the manuscript and tried to propose some measures to improve the prognosis of patients with cirrhosis. These were presented in the discussion chapter.

Reviewer 2 Report

Comments and Suggestions for Authors

The authors explore the relationship between thyroid hormone levels (T3, fT4, and TSH) and survival rates in patients with chronic liver disease. Increased TSH levels during hospitalization in patients with cirrhosis may be associated with mortality. TSH and MELD may be prognostic factors of mortality in patients with liver cirrhosis. The article is written well, but some points must be included. If these points are fixed, it can be published.

The introduction needs to be extended, and the statistical analysis needs to be explained more understandably.

I think the discussion and the results need to be extended. What could be the reason for the increase or decrease of those hormones, and what could be the mechanism? The reason should be written by comparing the literature. They are writing that it could be a prognostic factor for mortality. Could the survival rate be extended by supplementing the hormones to an average level?

Author Response

We appreciate you for your precious time in reviewing our paper and providing valuable comments. Thank you!

Comments 1 - The introduction needs to be extended, and the statistical analysis needs to be explained more understandably.

Response 1- It is a good suggestion,  so we expanded the field of introduction information and explained the statistical results more clearly in the discussion chapter.

Comments 2- I think the discussion and the results need to be extended. What could be the reason for the increase or decrease of those hormones, and what could be the mechanism? The reason should be written by comparing the literature.

Response 2- Thank you for pointing this out. We agree with this comment. Therefore, we have improved the discussion chapter with pertinent explanations and in comparison, with the existing data in the literature.

Comments 3- They are writing that it could be a prognostic factor for mortality. Could the survival rate be extended by supplementing the hormones to an average level?

Response 3- It is a good suggestion considering that the studies must have clinical applicability. Thus, in the conclusions chapter, I formulated some hypotheses that deserve to be taken into consideration in the management of patients with liver cirrhosis.

Reviewer 3 Report

Comments and Suggestions for Authors

This research article focuses on thyroid hormons as predictors of survival in patients with liver cirrhosis. In the introduction authors have nicely showed how liver function may have an effect on thyroid metabolism. However in the methodology section we lack information regarding the follow-up period. What is minimum floow up period and what is maximum? How many patients were lost to follow up? Were all hospitally treated? Were hormons tested in the same lab? The authors state paraclinical data regarding patients? What do you mean by that? Also in the data analysis section, the reader cannot understand the meaning of ggplot, ggubr, etc? What are these?? Why this is important? Is this some kind of software? Spearmant test was used and showed in results although in methodology it has not been stated…in the results section, authors state grades of encephalopathy, which also has not been described in methodology. TE as etiology has not been described, its a short term for? Why the authors did not use Kaplan-Meier analysis in terms of survival? The results are not clearly written. Discussion is poor.

Comments on the Quality of English Language

English is fine.

Author Response

Thank you for the professional review, it is improving my manuscript.

Comments 1 What is the minimum flow-up period and what is the maximum?

Response 1-Thank you for your observation. Patients were observed for the entire duration of hospitalization, with an interval between 1 and 41 days, (M=7.21, SD=6.73). We will update the participant descriptions in the document.

Comments 2 How many patients were lost to follow-up?

Response 2 -Thank you for your observation. Indeed, only imputation modalities were reported, and not the losses between hospital admission and discharge. For TSH, only one missing value was found at admission and 3 missing values ​​at discharge. For T3, 5 missing values ​​were observed, the same participants at admission and discharge. For FT4, 2 missing values were found at admission and 3 missing values ​​at discharge. K-nearest neighbor imputation algorithm was used, and the ​​​​​​​analysis was not a survival, but a comparison between means of continuous variables. Recorded hormone values ​​as they resulted from the analysis (continuous variables, higher power) were used instead of transformation to categorical values ​​(categorical variables, lower power), and statistical methods were used to compare rank means. There was no follow-up moment, data collection took place only at admission and discharge from the hospital. We will update the method section in the document and thank you for your observation.

Comments 3- Were all hospital treated?

Response 3 Yes, all participants were treated in the hospital. Thank you for your observation, and we will update the participant descriptions in the document.

Comments 4 Were hormones tested in the same lab?

Response 4 – The hormone tests were performed in the same lab, we added this mention in the manuscript.

Comments 5 The authors state paraclinical data regarding patients. What do you mean by that?

Response 5 We wanted to say that we used paraclinical data of the patients with their consent. We revised the expression, and the information is clearer now.

Comments 6- Also in the data analysis section, the reader cannot understand the meaning of ggplot, ggubr, etc? What are these?? Why this is important? Is this some kind of software?

Response 6 -As mentioned in the data analysis, processing was performed using the R language, and a series of specialized packages: ggplot, ggubr, etc. represent specialized packages with which we performed analyses and constructed graphs. Reporting practices require citing the applications in which the analyses were performed and their versions; thus, they were included in the article.

Spearmant test was used and showed in results although in the methodology it has not been stated…

Thank you for your observation. Indeed, it is an unfortunate omission. Spearman's rho correlation coefficient was used to analyze the association (and their significance) between the main variables. We will include the following in the methods section: "As the continuous variables analyzed did not have a univariate normal distribution, a Spearman correlation matrix will be used to investigate bivariate associations"

Comments 7- In the results section, authors state grades of encephalopathy, which also has not been described in the methodology.

Response 7- The observation is very good, we added the criteria (West Haven) which were the basis of the encephalopathy quantification and classification in the studied patients

Comments8- TE as etiology has not been described, it's a short-term for?

Response 8-- We made the change, it was about alcoholic etiology (toxic-ethanolic)

 Comments 9- Why did the authors not use Kaplan-Meier analysis in terms of survival?

Response - Thank you for your suggestion. The original study design was not intended for survival analysis; however, the suggestion is welcome. Being​​​​​​​ a non-parametric method aimed at the discretization of continuous hormonal values, we did not use it. We will update the article with this ​​​​​​​analysis as well, considering the golden rule regarding thyroid hormone values

Comments 10 - The results are not clearly written. Discussion is poor.

 Response 10 - Considering this observation, we revised the discussions for a better explanation of the results.

Round 2

Reviewer 1 Report

Comments and Suggestions for Authors

Dear authors,

Thank you for incorporating all my suggestions in the manuscript. It has now become a much better version of itself and I would like to congratulate you and suggest publishing it.